# Prevalence and Outcomes of COVID-19 among Hematology/Oncology Patients and Providers of a Community-Facing Health System during the B1.1.529 (“Omicron”) SARS-CoV-2 Variant Wave

**DOI:** 10.3390/cancers14194629

**Published:** 2022-09-23

**Authors:** Samuel A. Kareff, Aliya Khan, Priscila Barreto-Coelho, Sunil Girish Iyer, Brian Pico, Michele Stanchina, Giselle Dutcher, José Monteiro de Oliveira Novaes, Aparna Nallagangula, Gilberto Lopes

**Affiliations:** 1University of Miami Sylvester Comprehensive Cancer Center/Jackson Memorial Hospital, Miami, FL 33136, USA; 2Broward Health North, Pompano Beach, FL 33064, USA; 3Memorial Cancer Institute, Pembroke Pines, FL 33028, USA; 4Department of Medicine, Division of Solid Tumor Oncology, University Hospitals Seidman Cancer Center, Case Western Reserve University, Cleveland, OH 44106, USA; 5College of Medicine, The University of Arizona, Tucson, AZ 85721, USA; 6School of Medicine, University of Miami, Miller Miami, FL 33136, USA

**Keywords:** COVID-19, Omicron, health care disparities, community oncology, safety net

## Abstract

**Simple Summary:**

The COVID-19 pandemic continues, and cancer patients are at high risk for both contracting as well as dying from the infection. There is not as much data known about newer COVID-19 variants such as Omicron compared to earlier waves for patients with cancer. In this study, we retrospectively evaluated how COVID-19 positivity affected both patients and their providers in our community-facing cancer clinic. We found that 33.3% compared to 8.7% of cancer providers versus patients, respectively, tested positive for COVID-19 from December 2021 through April 2022 (*p* = 0.038). Furthermore, we saw that almost two-thirds of cancer patients experienced delays in receiving cancer treatments. Finally, over 10% of cancer patients (4 of 90) died during the Omicron wave. This study confirms that COVID-19 remains a formidable infection in terms of cancer patients’ treatment as well as livelihood, and continues to result in considerable health care disparities for disadvantaged populations.

**Abstract:**

(1) Background: the SARS-CoV-2 (COVID-19) pandemic continues, and patients actively receiving chemotherapy are known to be at enhanced risk for developing symptomatic disease with poorer outcomes. Our study evaluated the prevalence of COVID-19 among patients and providers of our community-facing county health system during the B1.1.529 (“Omicron”) COVID-19 variant wave. (2) Methods: We retrospectively analyzed patients that received care and clinical providers whom worked at the Jackson Memorial Hospital Hematology/Oncology clinic in Miami, Florida, USA, from 1 December 2021 through 30 April 2022. We assessed demographic variables and quality outcomes among patients. (3) Results: 1031 patients and 18 providers were retrospectively analyzed. 90 patients tested positive for COVID-19 (8.73%), while 6 providers tested positive (33.3%) (*p* = 0.038). There were 4 (10.3%) COVID-19-related deaths (and another outside our study timeframe) and 39 non-COVID-19-related deaths (89.7%) in the patient population (*p* = 0.77). COVID-19 accounted for 4.44% of our clinic’s total mortality, and delayed care in 64.4% of patients. (4) Conclusions: The prevalence of COVID-19 positivity in our patient cohort mirrored local, state, and national trends, however a statistically significant greater proportion of our providers tested positive. Almost two-thirds of patients experienced a cancer treatment delay, significantly impacting oncologic care.

## 1. Introduction

The SARS-CoV-2 (COVID-19) pandemic continues within the U.S. with an average of nearly 100,000 daily cases as well as 400 daily deaths at present [1]. Patients actively receiving chemotherapy and whom identify as racial and/or ethnic minorities are known to be at enhanced risk for symptomatic disease, inpatient admission, as well as death [2]. Moreover, delays in cancer-related treatment are common with COVID-19 positivity [3], though data related to current variants’ transmission remain scarce. Finally, there is a lack of literature dedicated to investigating disparities in COVID-19 prevalence between oncologic patients and providers, with the most comprehensive studies having reported prevalence estimates during the widespread transmission of previous variants during the pandemic [4]. 

With the expected persistence of the COVID-19 pandemic, we aimed to evaluate outcomes related to COVID-19 in a unique and underreported clinic population of the Jackson Health System (JHS). JHS is a non-profit, academic health system located in Miami-Dade County, Florida that is tax-assisted and serves county residents based on financial need. The JHS consists of seven inpatient hospitals as well as multiple outpatient and urgent care centers. Jackson Memorial Hospital (JMH) serves as the academic and logistical hub of the health system, operating the system’s only dedicated Hematology/Oncology outpatient clinic and infusion center.

Therefore, our retrospective study evaluated the prevalence of COVID-19 among patients and providers of our community-facing county health system during the B1.1.529 (“Omicron”) COVID-19 variant wave. Additionally, we assessed cancer-related outcomes, highlighting delays in cancer-related therapies, during the same timeframe. Finally, we reviewed the literature dedicated to this topic, emphasizing the implications for cancer patients and providers both locally and globally.

## 2. Materials and Methods

We retrospectively analyzed patients receiving cancer-directed therapies and clinical providers working at the Jackson Memorial Hospital (JMH) Hematology/Oncology clinic, a safety net hospital system for Miami-Dade County in Miami, FL, USA, from 1 December 2021 through 30 April 2022. As we were unable to obtain sequencing data for each COVID-19 specimen received, we utilized this specific timeframe in conjunction with epidemiologic trends to assume that all samples were concurrent with the rise and predominance of the original Omicron variant B1.1.529. Inclusion criteria therefore consisted of active receipt of cancer-related therapies and having tested positive for COVID-19 during the study period.

Next, we analyzed risk factors leading to COVID-19 positivity including, but not limited to, age, race, ethnicity, cancer diagnosis, history of previous COVID-19, and vaccination status. We then assessed clinical presentation of COVID-19, including symptomatic disease (defined as any symptoms including upper respiratory illness, shortness of breath, fever, etc.); treatment with advanced COVID-19 therapies such as oral or intravenous antivirals (e.g., molnupiravir or nirmatrelvir/ritonavir), monoclonal antibodies (e.g., casirivimab/imdevimab or sotrovimab), convalescent plasma containing COVID-19-specific antibodies, steroids (e.g., dexamethasone), or interleukin-6 inhibitors (e.g., tocilizumab); interactions with outpatient and/or inpatient services including emergency department (ED)/urgent care visits, inpatient and/or intensive care unit (ICU) admissions, and deaths from COVID-19. We finally quantified outcomes such as delay of at least 7 days’ duration in cancer-directed therapies including, but not limited to, chemotherapy, hormone therapy, and immunotherapy delivery; radiotherapy administration; as well as surgical oncologic procedures. 

These data were categorized by the co-authors via retrospective chart view, and analyzed using Microsoft Excel^®^ software (Version 2207; Redmond, Washington, DC, USA). *p*-values were calculated using Fisher’s exact tests and considered significant at the less than 0.05 level. This study was approved by the University of Miami Institutional Review Board and Jackson Health System Clinical Trials Office under eProst 20,211,244 on 20 January 2022. 

## 3. Results

1031 patients and 18 providers were retrospectively analyzed during the study timeframe. The date of COVID-19 positivity was collected by calendar week and compared to local, state, and national cohorts (Figure 1). 

90 patients tested positive for COVID-19 (8.73% of total patients), while 6 providers tested positive (33.3% of total providers) (*p* = 0.038) (Table 1). There were 4 (10.3%) COVID-19 related deaths (and another outside our study timeframe), and 39 non-COVID-19-related deaths (89.7%) in the patient population (*p* = 0.77). Nearly two-thirds of patients whom tested positive experienced a cancer treatment delay of at least 7 days’ duration (*n* = 58; 64.4%), with the most common delays as interruptions or stops of treatment regimens followed by initial diagnosis and in- or outpatient initiation of therapy.

In terms of demographics, patients who tested positive were 54.4% female (*n* = 49), 35.6% Black (*n* = 32), 66.7% Hispanic/Latinx (*n* = 60), and 1.11% South Asian (*n* = 1) (Table 2). Only 6.67% of patients had tested positive for COVID-19 previously (*n* = 6), and there was one confirmed second and third reinfection in our cohort. 48.9% of patients testing positive were considered unvaccinated (*n* = 44) while 8.89% were boosted (*n* = 8) (see Table 2 for vaccination classification). The five most common malignancies represented in our cohort included breast cancer (25.6%), lymphoma (Hodgkin and Non-Hodgkin) (15.6%), lung cancer (non-small cell and small cell) (13.3%), colorectal cancer (6.7%), and multiple myeloma (5.6%). 

Concerning COVID-19 clinical presentation, 74.4% (*n* = 67) of patients presented with symptomatic disease and 57.8% (*n* = 52) sought care at an ED/urgent care setting. 43.3% of patients (*n* = 39) were admitted to the hospital, 10.0% were admitted to the ICU (*n* = 9), and 27.8% (*n* = 25) received advanced therapeutics (Table 3). Receipt of advanced therapeutics was highly associated with increased rates of admission to both the hospital (OR 13.7; *p* < 0.0001) and ICU (OR7.53; *p* = 0.0026). However, receiving advanced therapeutics did not statistically impact mean days of cancer-related treatment delay (17 versus 20 days, *p* = 0.0901).

Regarding the patients whom died of COVID-19, two were aged in the sixth decade and one in the fifth. All three were male, and two were unvaccinated while one was under-vaccinated. Two presented directly to the ED with shortness of breath or wheezing; received remdesivir and dexamethasone (with the second also having received tocilizumab); died due to multisystem organ failure in the medical ICU after 8 or 9 days; and had diagnoses of extensive-stage small cell lung cancer or gastric adenocarcinoma with pulmonary metastases, respectively. The third patient tested positive after developing flu-like symptoms on the medical ward, had a primary diagnoses of metastatic castrate-resistant prostate cancer with an incidental diagnosis of metastatic papillary thyroid cancer during the same admission, and died after 30 days of treatment delay in an inpatient hospice unit. We are unable to report clinical details for the fourth patient due to state-level regulatory statutes. 

## 4. Discussion

The prevalence of COVID-19 positivity in our cohort during the initial Omicron wave mirrored local, state, and national trends (Figure 1), however a statistically significant greater proportion of our providers tested positive. There is limited research regarding the disparate COVID-19 prevalence between cancer patients and providers in the Omicron era of the pandemic. COVID-19 prevalence estimates for cancer patients and providers displayed wide ranges from 1.7–1.8% in a French cohort [4] to 6.8–28.1% in two Belgian cohorts [6] in mid-2020, even though disease was clinically worse in the early pandemic. In the Delta (B1.617.2) and Omicron variant waves, there are little data related to cancer provider prevalence. Most research focuses on the efficacy of vaccines and the difference in symptomatic disease among cancer patients with Omicron variants [7]. It has been hypothesized that patients with cancer may adhere to greater self-protection, such as social distancing measures, during treatment, thereby decreasing overall risk and prevalence of infection [8]. Additionally, health care exposures have been found to be among the leading drivers of risk for COVID-19 seropositivity among health care workers, even during the more recent COVID variant waves. [9]. The CDC has recently redefined areas of high community transmission of COVID-19 due to both new cases and new admissions per 100,000 people, as well as percent of staffed inpatient beds occupied by COVID-19 patients [10]. Our study population met the former criteria for 10 of the 21 weeks in our timeframe, reaffirming the risk that both cancer patients and providers may experience for COVID-19 positivity, despite the public health achievements of vaccine-induced immunity at the population level.

COVID-19 infection in our patient population conferred a highly symptomatic presentation of disease. COVID-19 positivity resulted in almost three-quarters of our cohort developing symptoms (74.4%), and over half (57.4%) of our patients presenting to an ED or urgent care to obtain treatment. These rates are significantly higher than overall population estimates, with over half of those having seroconverted during the initial Omicron wave stating that they were unaware of their recent infection [11]. As has been previously reported, COVID-19 in cancer patients continues to behave as more clinically aggressive disease. 

COVID-19 not only resulted in more symptoms and urgent care visits for our clinic population, but also a high degree of inpatient health care utilization. Just under half of our patients were admitted to the hospital (43.3%) for COVID-19 during our study timeframe. Unfortunately, even fewer patients (27.8%) received advanced therapeutics such as steroids, monoclonal antibodies, interleukin-6 inhibitors, convalescent plasma, and oral antivirals. This finding was especially surprising given the greater availability of advanced COVID therapeutics that have been widely available since the onset of the Omicron wave. However, the proportion of our patients who received such treatments resonates with those recorded in larger, multinational cohorts [12]. A lack of patient and provider knowledge about the availability of such therapeutics, as well as varying emphasis in public health messaging campaigns to both providers and the lay population, may be partially responsible for the lack of prescribing for these patients known to be at higher risk for poorer outcomes.

Importantly, nearly two-thirds (64.4%) of our patient cohort experienced a delay of at least 7 days’ duration in cancer-related treatment, including chemotherapy, hormone therapy, and immunotherapy delivery; radiotherapy administration; as well as surgical oncologic procedures. Cancer treatment delays are generally recognized to contribute to worse long-term clinical outcomes, such as increased tumor size in head and neck cancer [13] or decreased survival in various oncologic and hematologic malignancies [14]. Quality outcomes have also been adversely affected, with an increase in both general psychological distress as well as post-traumatic stress disorder in patients who experienced COVID-19-associated treatment delays [15]. Finally, systemic outcomes have also been impacted, with a systemic decrease in life-years gained and resource-adjusted life-years gained when accounting for delays in cancer diagnostics and surgery [16]. The tension between mitigating COVID-19 and maintaining cancer treatment schedules will likely remain as the pandemic continues.

COVID-19 accounted for over 10% of our clinic population’s total mortality during the study period, a prevalence approaching that of larger international cohorts during the first few months of the pandemic [17] and similar to single-center cohorts during the Omicron waves [18]. Accounting for an additional death attributable to COVID-19 after the study period terminated in May 2022, this percentage reaches 12.5% (*n* = 5/40). Such a high COVID-19 mortality rate emphasizes the importance for both oncology patients and providers in averting COVID-19 positivity, an increasingly difficult goal as the SARS-CoV-2 virus continues to evolve into more transmissible and potentially more vaccine-resistant variants. International cancer guidelines continue to emphasize reducing high-risk COVID-19 exposures in order to minimize the negative impact on cancer outcomes. Increasing telemedicine use, switching to oral (versus intravenous or subcutaneous) medication formulations, and administering supportive care in the outpatient setting are among some of the recommended strategies [19]. While these guidelines are based on the best available evidence, they can be logistically difficult to implement in under-served clinics such as ours represented in this study. Care delivery innovation remains key.

Patients of color remain at higher risk for both COVID-19 infection [20] and subsequent treatment delay [21], which held true for all 90 patients who tested positive in our cohort. Moreover, our cohort is unique among others represented in the literature as our catchment area includes one of two metropolitan areas in North America in which greater than 50% of residents are born outside the country, with over 90% originating from Latin America and the Caribbean in our county alone [22]. Immigrant populations are at heightened risk for exposure to COVID-19 for various reasons including lack of formal documentation, “frontline” occupations, overcrowded living conditions, and lack of access to health care [23]. Therefore, the results of our cohort may not be readily generalizable to other oncology populations, but do highlight some unique challenges our clinic faces.

Strengths of our study include the following: (1) the primary analysis of both cancer patients and their providers during the same timeframe, both of which have not been routinely studied significantly since the original COVID-19 waves in early to-mid-2020; (2) the unique population of our safety-net clinic cohort in a demographically diverse metropolitan area of the United States; and (3) the detailed information related to cancer-related outcomes including, but not limited to, severity of COVID-19, receipt of appropriate COVID-19-directed as well as cancer-related therapies; and deaths. Limitations of our study include its single-center nature as well as lack of comparison to earlier and evolving COVID-19 variant waves. Additionally, there may be a sample bias in evaluating cancer patients as symptoms similar to those of COVID-19 attributed to either cancer itself or its treatment may result in higher rates of testing and detection [24] As this was a retrospective chart review, it is also possible that the rates in our study may have underestimated asymptomatic cases in both patients and providers. Finally, in a few cases we were unable to report all relevant clinical factors related to our study’s aims given policy restrictions related to clinical research at our public institution. Such systems may look to simplify regulatory limitations in order to bolster research efforts moving forward.

## 5. Conclusions

Significant disparities in COVID-19 outcomes are displayed in our study. First, providers serving our safety-net, community-facing Hematology/Oncology clinic were significantly more likely than the patients whom the clinic served to test positive during the first Omicron wave. While individual-level behaviors may explain a degree of this divergence, persistent exposures in the health care space likely drive some proportion of the increased seropositivity among providers we detected during our study. Health care systems must ensure that not only the most vulnerable patients (such as those actively receiving cancer-directed therapies) but also the providers who serve them are adequately protected from excess exposures. Interventions such as mandatory use of (K)N95 respirators at all times within the health care physical environment can help decrease nosocomial transmission [25].

In our Hematology/Oncology clinic, nearly two-thirds of the who developed COVID-19 experienced delays in cancer-related treatment, including, but not limited to, chemotherapy, hormone therapy, or immunotherapy delivery; radiotherapy administration; and surgical oncologic procedures. Given the expected emergence of new Omicron sub-variants as well as persistence of the pandemic for the foreseeable future, health care systems will need to continue innovating care delivery methods that minimize delays in treatment and healthcare-related exposures. Such recommendations include incorporating telemedicine for routine outpatient monitoring and/or survivorship visits, increasing cycle timing of maintenance therapies (i.e., maintenance pembrolizumab immunotherapy in 6-week versus 3-week cycles), reducing radiotherapy fractionation without sacrificing clinical efficacy [26], using liquid rather than tissue biopsies for molecular analysis of late-line treatments [27], and leveraging additional cycles of chemotherapy as a stopgap when surgical oncology procedures are delayed due to overburdened operating room scheduling or COVID-19-positive staff [28].

Finally, our study demonstrated many of the well-documented sequelae of COVID-19 for some of the most vulnerable patients, including patients of color and immigrants. Systems that serve these disadvantaged populations often rely on limited public funding in order to treat both individuals and communities with multiple socioeconomic barriers to timely and cost-effective care. COVID-19 has become the third-leading cause of death in the U.S. during the pandemic [29], and it is estimated that a single-payer universal healthcare system would have saved both 212,000 lives as well as $105.6 billion in COVID-19-related morbidity and mortality in 2020 alone [30]. Therefore, the need for health care delivery and systems reform, underscoring intersectional populations such as that of our clinic, becomes even more crucial.

## Figures and Tables

**Figure 1 cancers-14-04629-f001:**
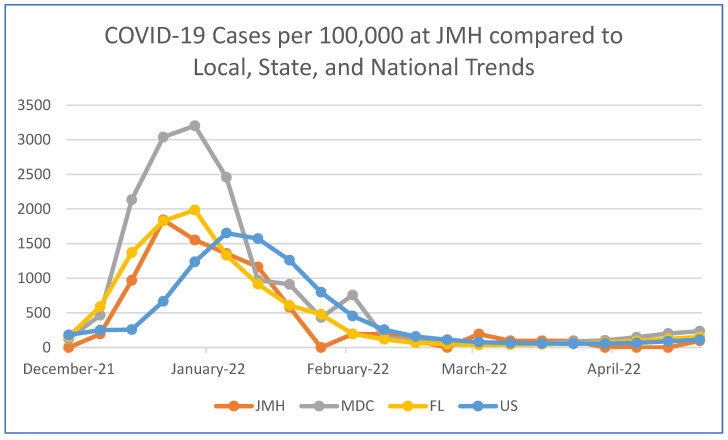
COVID-19 cases per 100,000 people in the JMH Hematology/Oncology clinic population compared to Miami-Dade County (MDC), the State of Florida (FL) [5], and the United States (US) [1].

**Table 1 cancers-14-04629-t001:** COVID-19 Positivity and Key Outcomes among Hematology/Oncology Patients and Providers during the Initial Omicron Wave.

Variable	*n* (% Total)	Variable	*n* (% Total)	*p*-Value *
*Total Providers*	18	*Total Patients*	1031	**0.038**
Providers testing positive for COVID-19	6 (33.3%)	Patients testing positive for COVID-19	90 (8.73%)	
Providers not testing positive for COVID-19	12 (66.7%)	Patients not testing positive for COVID-19	941 (91.27%)	
*Total Deaths*	39			0.77
Deaths due to COVID-19	4 (10.3%) ^1^			
Deaths due to Cancer or Other Causes	35 (89.7%)			
*Cancer Treatment Delays due to COVID-19*		*Median Days of Treatment Delay*		Not applicable
Yes	58 (64.4%)	20 days (range 7–65 days) (IQR 15)		
No	32 (35.6%)			

^1^ One COVID-19-related death occurred outside our study timeframe and is not included in this table. * *p*-values were calculated using Fisher’s exact test. This table lists various outcomes related to COVID-19 positivity among patients and providers of the JMH Hematology/Oncology clinic during the months of December 2021 through April 2022. Statistically significant outcomes are reported in bold.

**Table 2 cancers-14-04629-t002:** Demographics of Patients (*n* = 90) whom Tested Positive for COVID-19.

Demographic Category	Sub-Category	*n* (% Total)
Race	Black	32 (35.6)
	South Asian	1 (1.1)
	White	57 (63.3)
Ethnicity	Hispanic/Latinx	60 (66.7)
	Not Hispanic/Latinx	30 (33.3)
Gender	Female	49 (54.4)
	Male	41 (45.6)
Age	20–29	2 (2.2)
	30–39	9 (11.1)
	40–49	11 (12.2)
	50–59	31 (34.4)
	60–69	27 (30.0)
	70–79	7 (7.8)
	80–89	2 (2.2)
	90–99	1 (1.1)
Condition	Breast Cancer	23 (25.6%)
	Lymphoma	14 (15.6%)
	Lung Cancer	12 (13.3%)
	Colorectal Cancer	6 (6.7%)
	Multiple Myeloma	5 (5.6%)
	Prostate Cancer	4 (4.4%)
	Multiple Cancers	3 (3.3%)
	Testicular Cancer	2 (2.2%)
	Renal Cell Carcinoma	2 (2.2%)
	Stomach Cancer	2 (2.2%)
	Leukemia	2 (2.2%)
	Brain Cancer	2 (2.2%)
	Head and Neck Cancer	2 (2.2%)
	Other ^1^	11 (12.3%)
History of Previous COVID-19	Yes	6 (6.7)
	No	84 (93.3)
Vaccination Status	Unvaccinated	36 (40.0)
	Under-vaccinated ^2^	8 (8.9)
	Vaccinated ^3^	36 (40.0)
	Boosted ^4^	8 (8.9)
	Unknown	2 (2.2)

This table lists various demographic variables, including race, ethnicity, gender, age, condition, history of previous COVID-19 infection, and vaccination status, among patients whom tested positive for COVID-19 during the study timeframe. ^1^ Other included one case each of urothelial carcinoma, sarcoma, myeloproliferative neoplasm, dendritic cell neoplasm, Rosai-Dorfman disease, esophageal cancer, small bowel cancer, appendiceal carcinoma, cholangiocarcinoma, neuroendocrine tumor, and idiopathic thrombocytopenia purpura. ^2^ Under-vaccinated was considered to have received only 1 mRNA-1273 or BNT162b2 vaccine. ^3^ Vaccinated was considered to have received 2 total mRNA-1273 and/or BNT162b2 vaccines, or 1 Ad26.COV2.S vaccine. ^4^ Boosted was considered to have received 3 total mRNA-1273 and/or BNT162b2 vaccines, or 1 Ad26.COV2.S vaccine with a mRNA-1273 or BNT162b2 booster vaccine.

**Table 3 cancers-14-04629-t003:** Clinical Outcomes of Patients whom Tested Positive for COVID-19.

Clinical Outcome	Sub-Category	*n* (%)
Symptomatic Disease	Yes	67 (74.4)
	No	23 (25.6)
ED or Urgent Care Visit	Yes	52 (57.8)
	No	38 (42.2)
Admission to Hospital	Yes	39 (43.3)
	No	51 (56.7)
Admission to ICU	Yes	9 (10.0)
	No	81 (90.0)
Advanced Therapeutics	Yes	25 (27.8)
	No	65 (72.2)

This table lists various clinical outcomes, including symptomatic disease, ED or urgent care visit, admission to hospital, admission to ICU, and receipt of advanced therapeutics for patients whom tested positive for COVID-19.

## Data Availability

The data presented in this study may be available upon request from the corresponding author. The data are not publicly available due to limitations on data release pursuant to Data Exchange policy between the Public Health Trust of Miami-Dade County and the University of Miami.

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
