# Peer review of "Prevalence and Outcomes of COVID-19 among Hematology/Oncology Patients and Providers of a Community-Facing Health System during the B1.1.529 (“Omicron”) SARS-CoV-2 Variant Wave"

_cancers, 2022, doi:10.3390/cancers14194629_

Round 1

Reviewer 1 Report

This article presents unique data about the prevalence of SARS-CoV-2 positivity in hematooncology group of patients and healthcare providers during the Omicron wave of COVID-19. I believe this article will be helpful in the cancer field during C19-pandemic.
However, there are still some major comments that might make the article better:

1.      Methods and materials: there is a lack of information about the “scheme” of SARS-CoV-2 testing in patients and healthcare providers, especially. Was it routinely done in the healthcare providers group during the analyzed period (from Dec-21 to Apr-22)? Is it truly reflect the %  of the positive test results in the care providers group (also the “asymptomatic” cases)?

2.      Please, add the information on how many healthcare providers received COVID-19-directed therapy (“advanced therapeutics”) vs a group of cancer patients and comment on the obtained results.

3.      Would you like to add the comparison of the COVID-19 outcome in a group of cancer patients who received (“advanced therapeutics”) or not the COVID-19-directed treatment? Are there any differences in terms of admission to the hospital, admission to the ICU or postponement of the anticancer therapy? 

Author Response

RESPONSE TO REVIEWER 1 COMMENTS

This article presents unique data about the prevalence of SARS-CoV-2 positivity in hematooncology group of patients and healthcare providers during the Omicron wave of COVID-19. I believe this article will be helpful in the cancer field during C19-pandemic.

However, there are still some major comments that might make the article better:

  1. Methods and materials: there is a lack of information about the “scheme” of SARS-CoV-2 testing in patients and healthcare providers, especially. Was it routinely done in the healthcare providers group during the analyzed period (from Dec-21 to Apr-22)? Is it truly reflect the %  of the positive test results in the care providers group (also the “asymptomatic” cases)?

As this was a retrospective study utilizing chart review, we did not employ routine SARS-CoV-2 testing and were aware of any systems-based methodology to do so in patients or healthcare providers.  Symptoms were the most common trigger for both groups to be tested, especially for patients receiving their cancer-directed therapies.  As such, we agree with Reviewer 1 that it is possible that the rates detected in our study may underestimate asymptomatic cases.  We have added comments reflecting this on page 7, lines 41-42.

  1. Please, add the information on how many healthcare providers received COVID-19-directed therapy (“advanced therapeutics”) vs a group of cancer patients and comment on the obtained results.

While we agree with Reviewer 1 that this comparison would have added to the scientific value of this piece, our university-based IRB and hospital-affiliated Clinical Trials Office explicitly forbid this data collection so as to protect the privacy of the healthcare providers examined in our study. Please see the attached IRB approvals and protocol, as well as similar comment below under Reviewer 2 for more information.

  1. Would you like to add the comparison of the COVID-19 outcome in a group of cancer patients who received (“advanced therapeutics”) or not the COVID-19-directed treatment? Are there any differences in terms of admission to the hospital, admission to the ICU or postponement of the anticancer therapy?

We agree that this was an important comparison to add to our analysis.  We included any possible differences in terms of hospital (OR 13.7; p<0.0001) and ICU admission (OR 7.53; p=0.0026) in the forms of odds ratios in the Results on line 33-35, page 3. We also list the non-statistically significant 3-day change in median treatment-related delays on line 34-37, page 3. We had previously postulated some reasons for this disparity on line 34-43, page 6, especially having to do with access to care for this patient population.

We separately are obliged to report that we discovered two errors in our data reporting as a result of this comment due to data migration errors.  The first relates to raw count and percentages of patients whom received advanced therapeutics.  We had initially reported 67 patients did not receive advanced therapeutics and 23 had when, in reality, the correct numbers were 65 and 25, respectively (corrected in Table 3 as well as lines 31-32, page 3).  The second relates to the raw count and percentage of patients having multiple myeloma and multiple cancers as primary diagnoses.  We have respectively corrected these case counts and corresponding percentages in Table 2 as well as lines 28-29, page 3. We greatly apologize for these errors, and have double checked all our data to ensure that no additional errors were present after this peer review.

Reviewer 2 Report

The authors report on a single institution experience of exposure and outcomes of cancer patients and providers during the Omicron SARS-CoV-2 Variant Wave from 12/1/21-4/30/22. The results are important addition to the literature, as they show detailed outcomes of patients who tested positive for SARS-CoV-2 in a predominantly Black/Hispanic patient population, including disease positive rates, disease outcomes, and association of cancer treatment delay.

I ask the authors to comment on the following major considerations in hopes of strengthening the overall manuscript:

-          Table 1 – P value not clearly whether the comparison is between positive provider vs positive patient. It seems through the manuscript that the significance is in comparison to difference between positive vs negative provide and positive vs negative patient, would suggest reorganization of table to make more clear. .

-          Table 2 – would be more impactful to have demographic breakdown with total cohort (ie. n=1000) and then comparison of any differences in demographic categories between total population and COVID+ patients. Further, would suggest listing total population in top row (ie. n=90).

-          Could authors provide further description of 4 covid-related deaths, such COVID course/treatment, vaccine status, primary malignancy, amount of treatment delay, etc.

-          Authors describe the very significant endpoint of cancer treatment delay, however the overall definition is quite broad (a delay of at least 7 days’ duration in cancer-related treatment, including chemotherapy, hormone therapy, and immunotherapy delivery; radiotherapy administration; as well as surgical oncologic procedures). Great description would be more helpful, such as:

o   How many days of treatment delay total (such as median, range)

o   When during treatment were treatment delays most common (ie. initial treatment, stop in treatment, dose reduction, etc.)

-          Would suggest slight clarification in simple summary to clarify the 10% of patients who died were COVID-related of total deaths in cohort, or add that deaths within COVID-positive patients were 4/90.

-          Can authors comment on if there were any censoring of patients/providers during this time from the analysis, or were total patients that received cancer treatment during this time period were all assessed and included in analysis.

-          Can authors confirm that inclusion criteria stipulated patients received cancer therapy AND tested for COVID19 during study time frame of 12/1/21-4/30/22.

Author Response

RESPONSE TO REVIEWER 2 COMMENTS

The authors report on a single institution experience of exposure and outcomes of cancer patients and providers during the Omicron SARS-CoV-2 Variant Wave from 12/1/21-4/30/22. The results are important addition to the literature, as they show detailed outcomes of patients who tested positive for SARS-CoV-2 in a predominantly Black/Hispanic patient population, including disease positive rates, disease outcomes, and association of cancer treatment delay.

I ask the authors to comment on the following major considerations in hopes of strengthening the overall manuscript:

-          Table 1 – P value not clearly whether the comparison is between positive provider vs positive patient. It seems through the manuscript that the significance is in comparison to difference between positive vs negative provide and positive vs negative patient, would suggest reorganization of table to make more clear. .

             Thank you for this point of clarification.  We agree with Reviewer 2 and hope we have more clearly presented these data in our newly formatted Table 1 (page 4) which lists relevant p-values by row.

-          Table 2 – would be more impactful to have demographic breakdown with total cohort (ie. n=1000) and then comparison of any differences in demographic categories between total population and COVID+ patients. Further, would suggest listing total population in top row (ie. n=90).

             We agree that the data would be more impactful if we were able to present the demographic breakdown of our total clinic cohort.  Unfortunately, the reporting of specific data outside clinic patients whom tested positive for COVID-19 was explicitly forbidden by our university-based IRB and hospital-affiliated Clinical Trials Office (please see attached IRB approvals and protocol).

             Separately, we listed the total population in the top row of Table 2 as suggested (page 4,  line 11).

-          Could authors provide further description of 4 covid-related deaths, such COVID course/treatment, vaccine status, primary malignancy, amount of treatment delay, etc.

             Yes, we are glad to provide a further description of 3 of the 4 COVID-related deaths, and have included these data within the Results section on page 3, lines 38-47. We are unable to provide the fourth due to censorship as mandated in Florida Statue 945.10 (1), and have listed this rationale on line 47-48, page 3:

             1) 53-year-old man with COVID detected as an inpatient in the workup of an incidental diagnosis of metastatic papillary thyroid carcinoma. He received remdesivir and dexamethasone. He was unvaccinated and with a primary malignancy of metastatic castrate-resistant prostate cancer.  His treatment delay was 30 days, and he eventually died with inpatient hospice

             2) 58-year-old man whom presented to the ED with wheezing.  He received remdesivir and dexamethasone.  He was unvaccinated and with a primary diagnosis of extensive-stage small cell lung cancer.  His treatment delay was 9 days, and he eventually died due to multiorgan system failure in medical ICU.

             3) 49-year-old man whom presented to ED with shortness of breath.  He received tocilizumab, remdesivir and dexamethasone.  He received 1 dose of Pfizer (under-vaccinated).  His primary diagnosis was gastric cancer with pulmonary metastases.  His treatment delay was 8 days, and he eventually died due to multisystem organ failure in medical ICU.

-          Authors describe the very significant endpoint of cancer treatment delay, however the overall definition is quite broad (a delay of at least 7 days’ duration in cancer-related treatment, including chemotherapy, hormone therapy, and immunotherapy delivery; radiotherapy administration; as well as surgical oncologic procedures). Great description would be more helpful, such as:

  • How many days of treatment delay total (such as median, range)
    • We agree that displaying more descriptive statistics is helpful in understanding the impact of COVID-19 positivity and treatment delay. The total sample’s descriptive statistics included a median 20-day treatment delay with a range of 7-56 days and IQR 15. We included these data in lines 16-18, Page 3 as well as in Table 1.

  • When during treatment were treatment delays most common (ie. initial treatment, stop in treatment, dose reduction, etc.)
    • During treatment, delays were most common as interruptions or stops in treatment followed by initial diagnosis and in- or outpatient initiation of therapy. We mention this observation on line 18-20, page 3.

-          Would suggest slight clarification in simple summary to clarify the 10% of patients who died were COVID-related of total deaths in cohort, or add that deaths within COVID-positive patients were 4/90.

Thank you for this clarification.  We have adjusted it in the Simple Summary accordingly on line 24, page 1.

-          Can authors comment on if there were any censoring of patients/providers during this time from the analysis, or were total patients that received cancer treatment during this time period were all assessed and included in analysis.

Yes, we are obligated to again comment on the many legislative and policy restrictions employed in the clinical research that takes place at our institution as a publicly-funded health system. There are several restrictions employed by the Public Health Trust (funder of Jackson Health System) of Miami-Dade County in terms of conducting clinical research and reporting data.  Some of these restrictions include, but are not limited to, the dissemination of only the most strictly necessary data in order to inform clinical research, such as those presented in our study.  As such, we were unable to report any data related to providers in this study apart from known COVID-19 positivity, such as date of positivity, demographics of the providers, or equivalent outcomes such as ED/urgent care visit, hospital admission, and others.  Furthermore, as our health system serves especially vulnerable populations, such as prisoners, we are unable to disclose any potentially identifiable protected health information.  We hope these restrictions are understandable by the reviewers as a least a couple previous comments are unable to be addressed due to these restrictions.  We have previously included the IRB approval notifications which reference any applicable system-based regulations.

-          Can authors confirm that inclusion criteria stipulated patients received cancer therapy AND tested for COVID19 during study time frame of 12/1/21-4/30/22.

Yes, we can confirm these inclusion criteria.  We have restated them explicitly on line 35-37, page 2.

Reviewer 3 Report

Here, the authors have assessed the Prevalence and Outcomes of COVID-19 among Hematology/Oncology Patients and Providers of a Community-Facing Health System during the B1.1.529 (“Omicron”) SARS-CoV-2 Variant Wave. The study also identified Patients of color at higher risk for COVID-19 infection and subsequent treatment delay. 

Author Response

RESPONSE TO REVIEWER 3 COMMENTS

Here, the authors have assessed the Prevalence and Outcomes of COVID-19 among Hematology/Oncology Patients and Providers of a Community-Facing Health System during the B1.1.529 (“Omicron”) SARS-CoV-2 Variant Wave. The study also identified Patients of color at higher risk for COVID-19 infection and subsequent treatment delay.

               We thank Reviewer 3 for the concise summary of our manuscript.

Round 2

Reviewer 2 Report

I appreciate the author's attempts to respond to suggestions, and overall feel the manuscript is greatly strengthened. 

The many legislative and policy restrictions for reporting clinical research is understandable. I believe this limitation should be mentioned in the discussion of the paper, as it is important to consider in the context of their data that they are able to present, though does not weaken the overall impact of the paper. The author's could speak on a broader comment of how these limitations hinder potential future research in this important question in general.

Author Response

I appreciate the author's attempts to respond to suggestions, and overall feel the manuscript is greatly strengthened. 

We thank you again for your continuous review.

The many legislative and policy restrictions for reporting clinical research is understandable. I believe this limitation should be mentioned in the discussion of the paper, as it is important to consider in the context of their data that they are able to present, though does not weaken the overall impact of the paper. The author's could speak on a broader comment of how these limitations hinder potential future research in this important question in general

We appreciate your understanding related to our regulatory burden. We agree that this limitation should be explicitly mentioned, and we added additional thoughts inspired by your comment on page 7, lines 41-45.